# Mesenchymal Stromal Cell Exosomes Mediate M2-like Macrophage Polarization through CD73/Ecto-5′-Nucleotidase Activity

**DOI:** 10.3390/pharmaceutics15051489

**Published:** 2023-05-13

**Authors:** Kristeen Ye Wen Teo, Shipin Zhang, Jia Tong Loh, Ruenn Chai Lai, Hwee Weng Dennis Hey, Kong-Peng Lam, Sai Kiang Lim, Wei Seong Toh

**Affiliations:** 1Department of Orthopaedic Surgery, Yong Loo Lin School of Medicine, National University of Singapore, 1E Kent Ridge Road, Singapore 119228, Singapore; kristeenyewen@u.nus.edu (K.Y.W.T.);; 2Faculty of Dentistry, National University of Singapore, 9 Lower Kent Ridge Road, Singapore 119085, Singapore; 3Singapore Immunology Network, Agency for Science, Technology and Research (A*STAR), 8A Biomedical Grove, Singapore 138648, Singapore; 4Institute of Molecular and Cell Biology, Agency for Science, Technology and Research (A*STAR), 61 Biopolis Drive, Proteos, Singapore 138673, Singapore; lai_ruenn_chai@imcb.a-star.edu.sg (R.C.L.);; 5Tissue Engineering Program, Life Sciences Institute, National University of Singapore, 27 Medical Drive, Singapore 117510, Singapore; 6Integrative Sciences and Engineering Program, NUS Graduate School, National University of Singapore, 21 Lower Kent Ridge Road, Singapore 119077, Singapore

**Keywords:** mesenchymal stromal/stem cells, extracellular vesicles, exosomes, immunomodulation, macrophage, CD73

## Abstract

Mesenchymal stem/stromal cell (MSC) exosomes have been shown to alleviate immune dysfunction and inflammation in preclinical animal models. This therapeutic effect is attributed, in part, to their ability to promote the polarization of anti-inflammatory M2-like macrophages. One polarization mechanism has been shown to involve the activation of the MyD88-mediated toll-like receptor (TLR) signaling pathway by the presence of extra domain A-fibronectin (EDA-FN) within the MSC exosomes. Here, we uncovered an additional mechanism where MSC exosomes mediate M2-like macrophage polarization through exosomal CD73 activity. Specifically, we observed that polarization of M2-like macrophages by MSC exosomes was abolished in the presence of inhibitors of CD73 activity, adenosine receptors A_2A_ and A_2B_, and AKT/ERK phosphorylation. These findings suggest that MSC exosomes promote M2-like macrophage polarization by catalyzing the production of adenosine, which then binds to adenosine receptors A_2A_ and A_2B_ to activate AKT/ERK-dependent signaling pathways. Thus, CD73 represents an additional critical attribute of MSC exosomes in mediating M2-like macrophage polarization. These findings have implications for predicting the immunomodulatory potency of MSC exosome preparations.

## 1. Introduction

Immunomodulation plays an essential role in tissue repair and regeneration. Following injury, pro-inflammatory activities are necessary to neutralize injury and remove dead/injured tissue, while anti-inflammatory activities are important in facilitating migration and proliferation of reparative cell types to increase vascularization and nutrient supply needed for tissue repair and regeneration [1].

Macrophages are innate immune cells involved in host response to tissue injury [2]. Depending on the microenvironmental stimuli, naïve macrophages (M0) can be polarized into two distinct functional subsets, the pro-inflammatory M1 and anti-inflammatory M2 macrophages [3]. In general, inflammatory conditions such as interferon (IFN)-γ or lipopolysaccharide (LPS) polarize macrophages to the M1 phenotype, as characterized by the production of pro-inflammatory mediators such as tumor necrosis factor (TNF)-α, interleukin (IL)-1β, IL-6, nitric oxide (NO), and proteolytic enzymes, which aggravate inflammation and exacerbate tissue damage [4,5,6]. On the other hand, IL-4 or IL-13 can activate M2 macrophages to promote tissue homeostasis, resolution of inflammation, and tissue healing through the release of immunosuppressive molecules such as IL-10, transforming growth factor (TGF)-β, IL-1 receptor antagonist protein (IL-1RN), and arginase 1 (Arg1) [7,8,9]. These M2 macrophages are also characterized by the expression of CD163 and CD206 [9].

Mesenchymal stem/stromal cells (MSCs), which have well-documented immunomodulatory and regenerative properties, have been shown in many studies to promote M2-like over M1-like macrophage polarization [10,11,12]. The therapeutic efficacy of MSCs has been widely attributed to their secretion of extracellular vesicles (EVs) such as exosomes [13] and microvesicles [14], and indeed, accumulating evidence later establishes that they similarly possess immunomodulatory properties such as mediating M2-like macrophage polarization [15,16]. Macrophage polarization has been implicated as a key mechanism by which MSC exosomes alleviate disease severity in a range of preclinical models [17,18,19]. For instance, in a mouse model of hyperoxia-induced lung injury, MSC exosomes were observed to promote the infiltration of M2-like macrophages while simultaneously reducing the number of M1-like macrophages and pro-inflammatory cytokines such as TNF-α [17].

Despite the therapeutic potential of MSC exosomes, there are several challenges that need to be addressed. It is widely acknowledged that the therapeutic efficacy of MSC exosome preparation is influenced by the cell source, culture conditions, and isolation protocols. To overcome this and other limitations, immortalizing primary MSCs and establishing monoclonal MSC lines have been proposed and demonstrated as a feasible strategy to reduce the heterogeneity of MSCs and their derived exosomes [20]. We previously reported MYC transformation of human embryonic stem cell (ESC)-derived MSCs to generate a clonal E1-MYC 16.3 MSC line [20]. These immortalized E1-MYC MSCs grow faster and have increased telomerase activity while retaining the parental karyotype, thus providing an unlimited supply of MSCs for a scalable production of MSC exosomes in a consistent and reproducible manner [20]. These MSC exosomes are immunomodulatory and have been found to promote bone and cartilage repair by enhancing the infiltration of M2-like macrophages over M1-like macrophages with suppression of pro-inflammatory cytokines such as IL-1β and TNF-α [21,22]. In addition, they have been reported to enhance healing of radiation-induced injury by mobilizing monocytes from spleen and bone marrow to promote neovascularization at the wound site [23].

Although the mechanisms underlying these effects of MSC exosomes remain to be fully elucidated, MSC exosomes have been shown to induce an M2 phenotype in mouse and human monocytes through an extra domain A-fibronectin (EDA-FN) activated MyD88-dependent TLR signalling pathway [16]. As these MSC exosomes also carry several other immunomodulatory proteins [24,25], we investigate if these proteins could also mediate the polarization of M2 macrophages by MSC exosomes. A promising candidate was CD73, which is a surface ecto-5′-nucleotidase (NT5E) capable of converting adenosine monophosphate (AMP) to adenosine, and is widely recognized as an anti-inflammatory molecule, which exerts its effect through ubiquitously expressed receptors A_1_, A_2A_, A_2B_, and A_3_. In macrophages, adenosine has been shown to inhibit the production of pro-inflammatory cytokines such as TNF-α, IL-12, macrophage inflammatory protein (MIP)-2, and MIP-1α through interaction with A_2A_ [26,27] and A_2B_ adenosine receptors [28]. In other studies, adenosine reportedly enhanced IL-4 and IL-13 induction of M2 macrophage polarization through A_2B_ receptor and, to a lesser extent, A_2A_ receptor [29]. Consistent with these observations, activation of A_2B_ adenosine receptor was reported to augment macrophage production of IL-10 [30]. Based on these studies, we hypothesized that MSC exosomes derived from an immortalized E1-MYC 16.3 MSC line could also mediate M2-like macrophage polarization through its CD73/NT5E activity.

In this study, we show that MSC exosomes can directly polarize M0 macrophages to an anti-inflammatory M2-like phenotype via a CD73/adenosine-dependent mechanism. Specifically, exosomal CD73 mediated M2-like macrophage polarization through an AKT/ERK-dependent pathway, downstream of adenosine receptors A_2A_ and A_2B_. Our findings have provided a mechanistic context for MSC exosome-mediated polarization of M2-like macrophages, supporting the utility of MSC exosome therapy for tissue repair and regeneration.

## 2. Materials and Methods

### 2.1. Preparation and Characterization of MSC Exosomes

MSC exosomes were prepared from immortalized E1-MYC 16.3 human ESC-derived MSCs, as described previously [20]. Briefly, cells were cultured in Dulbecco’s Modified Eagle Medium (DMEM; Thermo Fisher Scientific, Waltham, MA, USA) with 10% fetal bovine serum (FBS; Thermo Fisher Scientific). For exosome preparation, cells were cultured in a chemically defined culture medium composed of DMEM supplemented with 1% nonessential amino acids, 1% glutamine, 1% insulin-transferrin-selenium-X, 1 mM sodium pyruvate, 0.05 mM β-mercaptoethanol, 5 ng/mL fibroblast growth factor (FGF)-2 (Thermo Fisher Scientific), and 5 ng/mL platelet-derived growth factor (PDGF)-AB (Peprotech, Cranbury, NJ, USA) [31]. After 3 days, the conditioned medium was size fractionated by tangential flow filtration and concentrated 50× using a membrane with a molecular weight cut-off of 100 kDa (Sartorius, Göttingen, Germany). The MSC exosome preparation was assayed for protein concentration using Coomassie Plus protein assay (Thermo Fisher Scientific), particle size distribution and concentration by ZetaView (Particle Metrix, Munich, Germany), and CD73/NT5E activity using the PiColorLock Gold Phosphate Detection System (Innova Biosciences, Cambridge, UK) in accordance with the identity and potency metrics proposed for MSC-sEV preparations [32,33]. This protocol for exosome preparation has been used for the preparation of more than 100 batches of exosomes with a high batch reproducibility in their protein and particle concentrations, modal diameter, and CD73/NT5E activity. For this study, batch AC113 was characterized to have protein concentration of 1.053 mg/mL, particle concentration of 1.33 × 10^11^ particles/mg, particle of a modal size of 139.29 nm, and CD73/NT5E activity of 22.71 ± 0.54 mU/μg.

### 2.2. Rat Primary Macrophage Culture

All procedures were performed according to guidelines stipulated by the Institutional Animal Care and Use Committee at National University of Singapore under protocol number: R18-1295. Briefly, peripheral blood mononuclear cells (PBMCs) were isolated from the blood of female 8-week-old Sprague Dawley rats by separation through Ficoll-Paque (Cytiva, Marlborough, MA, USA) density gradient centrifugation according to the manufacturer’s protocol. The isolated PBMCs were then seeded at density of 0.5 × 10^6^ cells/mL in Roswell Park Memorial Institute (RPMI) medium (Cytiva) supplemented with 1% (*v*/*v*) penicillin-streptomycin (PS; Thermo Fisher Scientific) and 10% (*v*/*v*) heat-inactivated FBS (Thermo Fisher Scientific). After overnight incubation, unattached cells were rinsed off with PBS, and the remaining cells were cultured in RPMI supplemented with 40 ng/mL rat macrophage colony-stimulating factor (M-CSF; Peprotech) for macrophage differentiation. Media change was performed every 2–3 days. Upon cell confluency >80% after 7–9 days of differentiation, macrophages were used for subsequent experiments.

### 2.3. Exosome Treatment and Inhibitor Study

To investigate the modulatory effects of MSC exosomes on macrophage polarization, the rat primary macrophages were treated with either 10 μg/mL exosomes or PBS vehicle for 24 and 48 h (Figure 1A). To investigate the role of CD73 in mediating the exosome effects, macrophages were co-treated with 10 μg/mL exosomes and 10 nM PSB12379 (CD73 inhibitor; Tocris Bioscience, Bristol, UK) (Figure 1B). The role of adenosine receptors in the activation of AKT and ERK pathways was investigated by pre-treating the cells with 2 μM of ZM241385 (A_2A_ receptor antagonist; Tocris), 200 nM PSB1115 (A_2B_ receptor antagonist; Tocris), 1 μM wortmannin (AKT inhibitor; Cell Signaling Technology, Danvers, MA, USA), or 10 μM U0126 (ERK inhibitor; Cell Signaling Technology) for 1 h, before treatment with 10 μg/mL exosomes (Figure 1C). For the inhibitor studies, the macrophages were harvested at 30 min for western blot analysis, 24 and 48 h for gene expression analysis, and 48 h for immunofluorescence staining (Figure 1B,C). All in vitro experiments were performed in triplicates (*n =* 3) in two independent trials.

### 2.4. Immunofluorescence Staining

The primary macrophages were washed with PBS and fixed with 4% paraformaldehyde (PFA; Sigma−Aldrich, St. Louis, MO, USA) for 30 min before permeabilization with 0.3% Triton X-100 (Sigma−Aldrich) for 15 min and blocking for 2 h with PBS supplemented with 10% FBS and 0.1% Triton X-100. Cells were then incubated overnight at 4 °C with anti-CD68 (1:100; Bio-Rad Laboratories, Hercules, CA, USA), anti-iNOS (1:200; Novus Biologicals, Centennial, CO, USA) and anti-CD206 (1:200; Abcam, Cambridge, UK) antibodies. Cells were then washed with PBS before incubation with the respective secondary antibodies, Alexa Fluor 594 goat anti-mouse IgG (1:1000; Thermo Fisher Scientific) or Alexa Fluor 488 goat anti-rabbit IgG (1:500; Thermo Fisher Scientific) antibodies for 2 h at room temperature. Finally, macrophages were counterstained with 4′,6-diamidino-2-phenylindole (DAPI) and examined under a fluorescence microscope (IX70, Olympus, Tokyo, Japan). The polarization of macrophages was analyzed by measuring the staining intensity in five randomly selected fields at 200× magnification and expressed as relative mean fluorescence intensity (MFI).

### 2.5. Quantitative Reverse Transcription Polymerase Chain Reaction

Quantitative reverse transcription polymerase chain reaction (qPCR) was used to analyze the gene expression of M1 and M2 associated markers. Total RNA from primary macrophages were isolated using PureLink^®^ RNA Mini kit (Thermo Fisher Scientific) according to manufacturer’s instruction. RNA from each sample was then reverse transcribed using iScript^TM^ Reverse Transcription Supermix (Bio-Rad) before amplification using CFX Connect^TM^ real-time PCR system (Bio-Rad) with iTaq^TM^ Universal SYBR^®^ Green Supermix (Bio-Rad), and primers as shown in Table 1. PCR cycling condition comprised initial denaturation at 95 °C for 30 s, followed by 40 cycles of amplification consisting of 15 s denaturation at 95 °C and 30 s extension at 60 °C. The mRNA expression levels were normalized against glyceraldehyde 3-phosphate dehydrogenase (*GAPDH*), calculated using comparative ∆CT method [34], and expressed as fold changes.

### 2.6. Western Blot Analysis

Western blotting was performed following a standard procedure [21]. Briefly, proteins were denatured, separated on 4–12% polyacrylamide gels (Thermo Fisher Scientific), blotted onto nitrocellulose membrane (GE Healthcare, Chicago, IL, USA), and probed with primary antibody followed by incubation with horseradish peroxidase (HRP)-coupled secondary antibody against the primary antibody. Table 2 shows the list of antibodies. After incubation, the protein bands were visualized using the WesternBright ECL HRP substrate (Advansta, San Jose, CA, USA) and then documented using the ChemiDoc™ MP System (Bio-Rad). The individual blots for all biological replicates are included in the Appendix A.

### 2.7. Statistical Analysis

Statistical analysis was performed using SPSS version 22.0 (SPSS, Chicago, IL, USA). The data were reported as mean ± SD and tested for normality. Statistical differences between the groups were determined by student’s *t* test or one-way ANOVA followed by Scheffe post hoc test for normally distributed data and Mann-Whitney U test or Kruskal–Wallis test followed by Dunn–Bonferroni post hoc test for non-normally distributed data. The statistical significance was set as *p* < 0.05.

## 3. Results

### 3.1. MSC Exosomes Promote M2-like Macrophage Polarization

To determine if MSC exosomes modulate macrophage polarization, we performed in vitro cell culture studies with primary naïve M0 macrophages differentiated from rat PBMCs using M-CSF. We first confirmed the expression of macrophage marker CD68 on PBMC-generated macrophages via immunofluorescence (Figure 2A,B). We observed that following exosome treatment, the expression of CD206, a M2 marker, was significantly elevated by ~3-folds (*p* < 0.001) on the CD68^+^ macrophages (Figure 2A,B). Enhanced M2 polarization of these macrophages was also confirmed by their M2-associated gene signature, including the upregulation of *Arg1*, *CCL24*, *CCR1*, *CD206, IL-1RN*, *IL-10*, *IL-13*, and *PPARγ*, as early as 24 h following exosome treatment (Figure 2C). However, MSC exosomes did not polarize macrophages toward a M1-like phenotype, as there were no changes in expression of M1 marker, iNOS, or other M1 signature genes such as *CCL5*, *CD80*, *IFN-γ*, *IL-1β*, *IL-12β*, *IL-6*, and *TNF-α* by the exosome-treated macrophages (Figure 2D–F). Collectively, these findings suggest that MSC exosomes can directly polarize M0 macrophages towards an anti-inflammatory M2-like phenotype.

### 3.2. MSC Exosomes Mediate M2-like Macrophage Polarization through CD73

We previously reported that MSC exosomes express CD73, a surface ecto-5′-nucleotidase (NT5E) that converts adenosine monophosphate (AMP) to adenosine [35], which in turn binds to the adenosine receptors to activate the AKT and ERK signaling pathways [36]. Given the well-documented immunomodulatory properties of adenosine [26,27,28,29,30,37], we hypothesize that MSC exosomes mediate M2-like macrophage polarization via a CD73/adenosine-dependent mechanism.

To test our hypothesis, we first confirm the presence of CD73 in MSC exosomes by western blot using anti-CD73 antibody (Figure 3A). We further demonstrated that the enzymatic activity of exosomal CD73 can be attenuated by 37.6 ± 7.2% with a specific CD73 inhibitor, PSB12379 (Figure 3B). To address the significance of exosomal CD73 on macrophage polarization, we treated M0 macrophages with MSC exosomes in the presence of PSB12379. We observed that CD73 inhibition by PSB12379 suppressed exosome-mediated M2-like macrophage polarization, as evidenced by a statistically significant reduction of ~62% in CD206 expression in the M0 + Exo + PSB12379 group, as compared to the M0 + Exo group (0.8 ± 0.2 vs. 2.1 ± 0.3, *p* = 0.001) (Figure 3C,D). In addition, gene expression analysis revealed that exosome-mediated upregulation of M2 signature genes such as A*rg1*, *CCR1*, *CD206*, *IL-1RN*, *IL-10*, and *PPARγ* were abrogated in the presence of CD73 inhibitor (Figure 3F). On the other hand, no induction of M1-like macrophages, as indicated by iNOS expression, was observed upon treatment with MSC exosomes in the presence or absence of CD73 inhibitor (Figure 3E). Collectively, these findings suggest that exosomal CD73 is required for M2-like macrophage polarization.

### 3.3. MSC Exosomes Promote M2-like Macrophage Polarization through Exosomal CD73-Mediated A_2A_ and A_2B_ Receptor Signaling

Next, to test whether exosomal CD73 mediates M2-like macrophage polarization through the adenosine pathway, we first investigated the AKT/ERK signaling downstream of adenosine receptors. Indeed, M0 + Exo macrophages show marked increase in the phosphorylation of AKT (1.3 ± 0.1 vs. 0.8 ± 0.03, *p* < 0.001) and ERK (3.0 ± 0.3 vs. 1.9 ± 0.3, *p* = 0.009) as compared to M0 macrophages (Figure 4A,B). This exosome-induced AKT and ERK phosphorylation was diminished in the presence of CD73 inhibitor with M0 + Exo + PSB12379 group having reduced phosphorylation of AKT (1.0 ± 0.1 vs. 1.3 ± 0.1, *p* = 0.003) and ERK (2.1 ± 0.2 vs. 3.0 ± 0.3, *p* = 0.026) as compared to the M0 + Exo group.

To confirm that exosome-mediated AKT and ERK phosphorylation occurs through adenosine signaling, we treated the cells with selective adenosine receptor A_2A_ and A_2B_ inhibitors, ZM241385 and PSB1115, respectively. Inhibition of A_2A_ receptor with ZM241385 suppressed exosome-induced AKT (1.2 ± 0.5 vs. 2.9 ± 0.7, *p* = 0.017) and ERK (2.8 ± 0.2 vs. 5.8 ± 0.6, *p* = 0.001) phosphorylation, while inhibition of A_2B_ receptor with PSB1115 resulted in the suppression of exosome-induced phosphorylation of AKT (0.9 ± 0.4 vs. 2.9 ± 0.7, *p* = 0.009), but not ERK (5.9 ± 0.7 vs. 5.8 ± 0.6, *p* = 0.991) (Figure 4C,D). We further demonstrated that inhibition of A_2A_ or A_2B_ receptor can significantly inhibit M2-like macrophage polarization, as evidenced by a statistically significant reduction in CD206 expression by ~59% (*p* < 0.001) and ~82% (*p* < 0.001), respectively (Figure 5A,B). On the contrary, inhibition of A_2A_ or A_2B_ receptor had no effect on M1-like macrophage polarization, as evidenced by no changes in iNOS expression across the groups (Figure 5C). Consistent with the decrease in CD206 expression by inhibition of A_2A_ or A_2B_ receptor, our gene expression results also showed that with the inhibition of A_2A_ receptor, there was significant downregulation of M2-associated genes such as *Arg1, CCR1, CD206, IL-10*, and *PPARγ* (Figure 5D), and, with inhibition of A_2B_ receptor, additional M2-associated genes including *CCL24, IL-1RN* and *IL-13* were downregulated (Figure 5D). Taken together, our findings suggest that MSC exosomes promote M2-like macrophage polarization through exosomal CD73-mediated A_2A_ and A_2B_ receptor activation.

### 3.4. Inhibition of AKT and ERK Attenuate MSC Exosome-Induced M2-like Macrophage Polarization

Finally, to establish the roles of AKT and ERK signaling in MSC exosome-induced M2-like macrophage polarization, we treated M0 macrophages with MSC exosomes in the presence of AKT and ERK inhibitors, wortmannin and U0126, respectively. As shown in Figure 6A, wortmannin significantly reduced exosome-induced AKT phosphorylation (0.3 ± 0.1 vs. 5.7 ± 0.5, *p* < 0.001) to a level comparable to the M0 control (0.3 ± 0.1 vs. 1.1 ± 0.4, *p* > 0.05). Wortmannin also reduced exosome-induced ERK phosphorylation, although the reduction was not statistically significant (2.8 ± 0.2 vs. 3.8 ± 0.8, *p* = 0.145). On the other hand, U0126 significantly reduced exosome-induced ERK phosphorylation (0.3 ± 0.1 vs. 3.8 ± 0.8, *p* < 0.001) to a level lower than that in M0 control (0.3 ± 0.1 vs. 2.4 ± 0.3, *p* = 0.003) (Figure 6A, B). However, U0126 did not reduce exosome-induced AKT phosphorylation as evidenced by comparable levels of AKT phosphorylation with or without ERK inhibition (4.4 ± 1.2 vs. 5.7 ± 0.5, *p* = 0.226) (Figure 6A,B).

Functionally, treatment with either wortmannin or U0126 abolished exosome-induced CD206 expression (0.5 ± 0.1 vs. 2.1 ± 0.2, *p* < 0.001) or (0.5 ± 0.2 vs. 2.1 ± 0.2, *p* < 0.001), respectively. The level of CD206 expression in both treatment groups was similar to the baseline expression level in M0 control (0.6 ± 0.1, *p* > 0.05), suggesting that the silencing of either AKT or ERK abrogated exosome-mediated M2 polarization (Figure 6C,D). Consistent with this, AKT or ERK inhibition also inhibited the expression of M2 signature genes such as *Arg1*, *CCL24*, *CCR1*, *CD206*, *IL-1RN*, *IL-10*, *IL-13*, and *PPARγ* (Figure 6F). Hence, these findings suggest that MSC exosomes promote M2-like macrophage polarization through AKT and ERK activation. Incidentally, AKT or ERK inhibition had no effect on M1 polarization, as evidenced by minimal changes in iNOS expression across the groups (Figure 6E).

## 4. Discussion

Several studies have reported that native MSC exosomes have the capacity to modulate immune cell signaling and responses [16,17,21,38]. Among the wide range of immune cell activities, MSC exosomes have been reported to enhance M2-like over M1-like macrophage polarization and infiltration, thereby suppressing inflammation to promote tissue repair in many animal disease models [17,21]. In this study, we demonstrated that CD73 on MSC exosomes promotes M2-like macrophage polarization through an AKT/ERK-dependent pathway, downstream of adenosine receptors, A_2A_ and A_2B_ (Figure 7). Our results have provided a mechanistic basis of how MSC exosomes could suppress inflammation to promote tissue repair and regeneration following injury or disease. This, together with our previous study demonstrating that MSC exosomes induced M2-like macrophage polarization by activating an EDA-FN-dependent MyD88-mediated TLR signaling [16], highlights a robust redundancy in MSC exosomes to promote M2-like macrophage polarization.

In this study, we hypothesized that in addition to EDA-FN, MSC exosomes could mediate M2-like macrophage polarization through CD73 activity. We have previously reported that MSC exosomes express CD73, a surface ecto-5′-nucleotidase (NT5E) [35], which converts AMP to adenosine to elicit diverse cellular signaling and responses through adenosine receptors, A_1_, A_2A_, A_2B_, and A_3_ [35,36,39]. By inhibiting CD73 with PSB12379, A_2A_ receptor with ZM241385, A_2B_ receptor with PSB1115, and AKT phosphorylation with wortmannin or ERK phosphorylation with U0126, we systematically defined and established the role of each protein in MSC exosome-mediated M2 macrophage polarization. We determined that MSC exosome-mediated M2 macrophage polarization by catalyzing the production of adenosine from AMP through CD73 expressed on the surface of MSC exosome. The adenosine binds A_2A_/A_2B_ adenosine receptor to activate AKT and ERK signaling pathways. Consistent with our observations, several studies have reported that adenosine being anti-inflammatory inhibits the production of pro-inflammatory cytokines by macrophages through interaction with the A_2A_ receptor [26,27] and A_2B_ receptor [28]. In other studies, adenosine reportedly enhanced IL-4 and IL-13 induced M2 macrophage polarization through A_2B_ receptor, and A_2A_ receptor at a lesser extent [29]. Additionally, activation of A_2B_ receptor was reported to augment the macrophage production of anti-inflammatory cytokine IL-10 [30]. Our results showed that whilst A_2A_ receptor triggered both AKT and ERK phosphorylation, and A_2B_ receptor induced mainly AKT phosphorylation, inhibition of either receptor led to a reduction of M2 macrophage polarization. However, it is important to note that adenosine receptors, which are G protein-coupled receptors (GPCR) have the potential to activate multiple signaling pathways beyond AKT and ERK. While A_1_ and A_3_ receptors couple with G protein subtype Gi to inhibit adenylate cyclase and reduce cyclic AMP (cAMP) production, A_2A_ and A_2B_ receptors bind to G protein subtype Gs and activate adenylate cyclase, resulting in increased cAMP production [40]. The elevated cAMP level was reported to promote M2 macrophage polarization through activation of PKA-mediated signaling pathways such as CREB [41] and STAT3 [42]. Therefore, there could be other adenosine receptor-responsive pathways mediating the effects of MSC exosomes in M2-like macrophage polarization.

We further confirmed that AKT and ERK signaling pathways were involved in exosome-mediated M2 macrophage polarization, as inhibition of exosome-induced AKT or ERK phosphorylation suppressed M2 macrophage polarization and their expression of M2-associated genes including *Arg1*, *CCL24*, *CCR1*, *CD206*, *IL-1RN*, *IL-10*, *IL-13*, and *PPARγ*. In agreement with our findings, several studies have reported that AKT phosphorylation is required for M2 macrophage polarization [43,44,45,46]. For instance, treatment with bone morphogenetic protein (BMP)-7 was shown to promote polarization of human monocytes into M2 macrophages through activation of the AKT signaling pathway [45]. Similarly, TGF-β was found to contribute to IL-4-induced M2 polarization through activation of AKT signaling [46]. Besides AKT, activation of ERK signaling was also reported to be involved in M2 macrophage polarization. For example, treatment of human monocytes with tumor-derived lactate resulted in the activation of ERK signaling, leading to M2-like macrophage differentiation [47]. Likewise, treatment of human monocytes with programmed death-ligand (PDL) 1 protein promoted M2 macrophage polarization through activation of ERK signaling pathway [48].

The functional effects of MSC exosomes have been largely attributed to their rich diverse cargo of proteins, lipids, nucleic acids, and metabolites [24,49]. Several studies have suggested that MSC exosome-induced macrophage polarization is associated with its cargo of miRNAs. For instance, it was reported that exosomes derived from mouse MSCs induced M2 phenotype of RAW264.7 cells through transfer of exosomal miR-21-5p [50]. Similarly, exosomes from human umbilical cord-derived MSCs were reported to attenuate mouse myocardial infarction injury by enhancing M2 macrophage polarization through exosomal miR-24-3p-mediated Plcb3/NF-κB signaling pathway [51]. However, accumulating evidence suggests that miRNAs in the exosome cargo may not be present in biologically relevant concentration and/or equipped with the biochemical functionality and potential required to elicit an appropriate timely biochemical response [52,53,54]. Our study therefore supports our previous hypothesis that the mechanism of action for MSC exosomes involves proteins, rather than miRNAs [52]. This is likely due to the low copy number of miRNAs in exosomes, which may not be sufficient to elicit significant biological effects [53,54]. Furthermore, recent studies have shown that the uptake of exosomes by cells is inefficient, and only a small fraction of internalized exosomes was able to escape lysosomal degradation [55,56]. As such, any hypothesis involving cellular uptake of exosomes as part of the mode of action is increasingly untenable.

In summary, our study demonstrates that MSC exosomes can effectively polarize M0 macrophages towards an anti-inflammatory M2-like phenotype through a CD73/adenosine-dependent mechanism. Specifically, we found that exosomal CD73 activates an AKT/ERK-dependent pathway downstream of adenosine receptors A_2A_ and A_2B_, leading to M2-like macrophage polarization. Our findings highlight the importance of CD73 as a key attribute of MSC exosomes responsible for inducing M2-like macrophage polarization, and suggest that it could be a valuable predictor of the immunomodulatory potency of MSC exosome preparations [33].

## Figures and Tables

**Figure 1 pharmaceutics-15-01489-f001:**
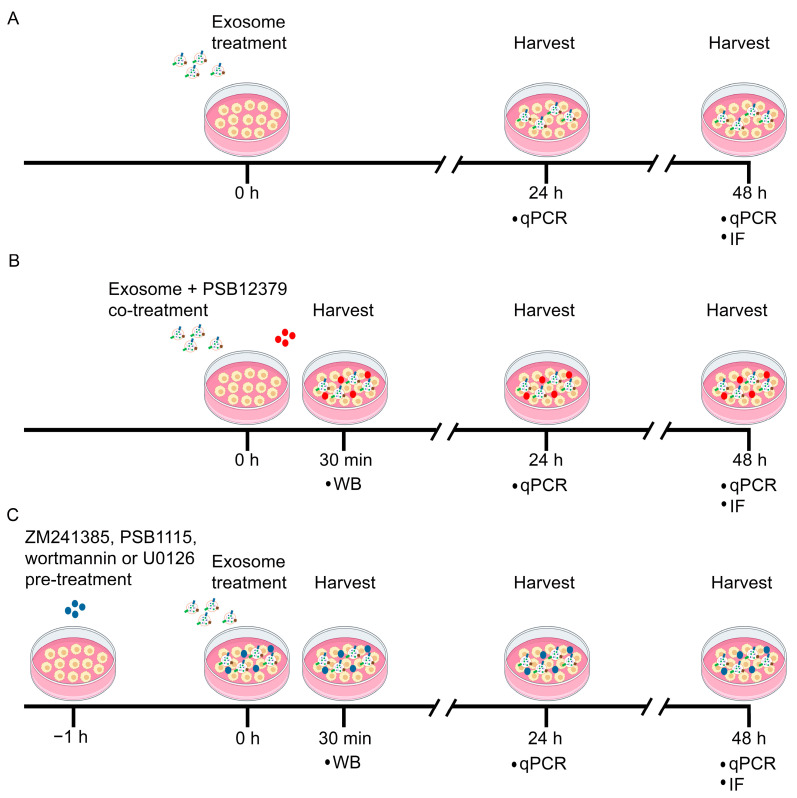
Time schedule for treatment of primary macrophages with MSC exosomes and inhibitors in vitro. (**A**) Primary macrophages were treated with MSC exosomes and harvested at 24 and 48 h for qPCR and IF. (**B**) Primary macrophages were co-treated with MSC exosomes and PSB12379 (CD73 inhibitor), before being harvested at 30 min for WB, and at 24 and 48 h for qPCR and IF analyses. (**C**) Primary macrophages were pre-treated with ZM241385 (A_2A_ receptor inhibitor), PSB1115 (A_2B_ receptor inhibitor), wortmannin (AKT inhibitor) or U0126 (ERK inhibitor) prior to exosome treatment, before being harvested at 30 min for WB, and at 24 and 48 h for qPCR and IF analyses. WB, western blot; qPCR, quantitative reverse transcription polymerase chain reaction; IF, immunofluorescence.

**Figure 2 pharmaceutics-15-01489-f002:**
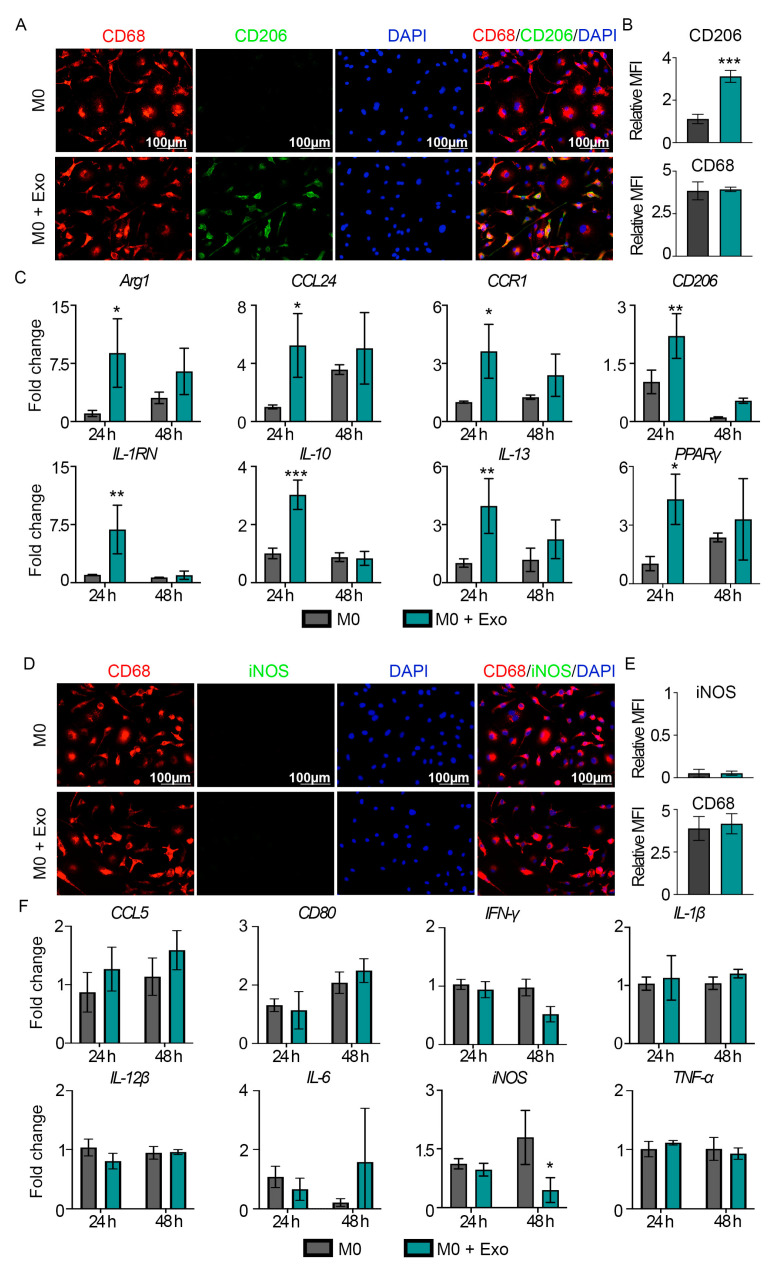
MSC exosomes promote M2-like but not M1-like macrophage polarization. (**A**) IF staining for CD206 and CD68 at 48 h. Representative images (*n =* 3). Scale bar: 100 μm. (**B**) Quantitative MFI of CD206 and CD68 at 48 h. (**C**) Gene expression analysis of M2-associated genes at 24 and 48 h. (**D**) IF staining for iNOS and CD68 at 48 h. Representative images (*n =* 3). Scale bar: 100 μm. (**E**) Quantitative MFI of iNOS and CD68 at 48 h. (**F**) Gene expression analysis of M1-associated genes at 24 and 48 h. Data are presented as mean ± SD. * *p* < 0.05, ** *p* < 0.01, and *** *p* < 0.001 compared to M0 macrophages. *n* = 3/group.

**Figure 3 pharmaceutics-15-01489-f003:**
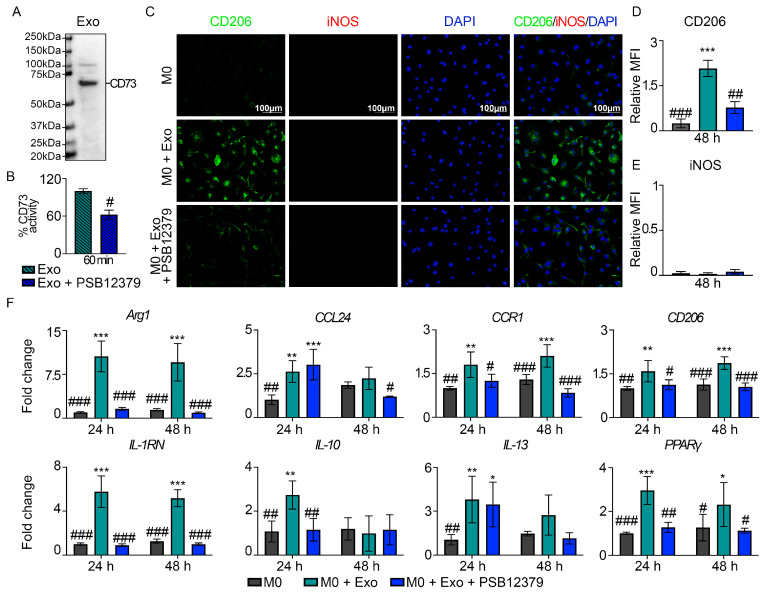
Exosomal CD73 mediates the effects of MSC exosomes in M2-like macrophage polarization. (**A**) Presence of CD73 protein band at ~70 kDa. (**B**) Percentage of CD73 activity in MSC exosomes following treatment with CD73 inhibitor (PSB12379) for 60 min. (**C**) IF staining of CD206 and iNOS in primary macrophages treated with MSC exosomes in presence or absence of CD73 inhibition (PSB12379) for 48 h. Representative images (*n =* 3). Scale bar: 100 μm. Quantitative MFI of (**D**) CD206 and (**E**) iNOS at 48 h. (**F**) Gene expression analysis of M2-associated genes at 24 and 48 h. Data are presented as mean ± SD. * *p* < 0.05, ** *p* < 0.01, *** *p* < 0.001 compared to M0, ^#^ *p* < 0.05, ^##^ *p* < 0.01, and ^###^ *p* < 0.001 compared to Exo or M0 + Exo group. *n =* 3–4/group.

**Figure 4 pharmaceutics-15-01489-f004:**
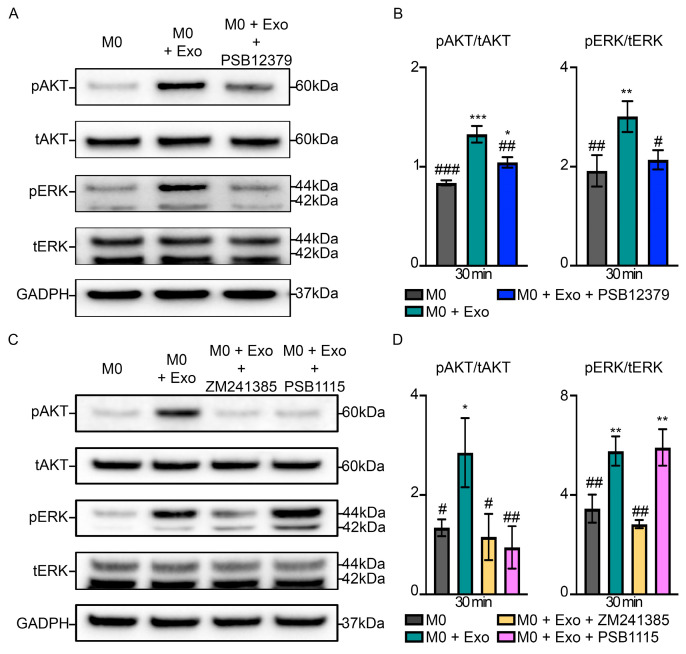
Exosomal CD73 mediates activation of AKT and ERK signaling through specific adenosine receptors A_2A_ and A_2B_. (**A**) Western blotting and (**B**) semi-quantitative analysis of AKT and ERK phosphorylation in primary macrophages treated with MSC exosomes in the presence or absence of CD73 inhibition by PSB12379. Representative images (*n =* 3). (**C**) Western blotting and (**D**) semi-quantitative analysis of AKT and ERK phosphorylation in primary macrophages pre-treated with A_2A_ receptor inhibitor (ZM241385) and A_2B_ receptor inhibitor (PSB1115) before exosome treatment. Representative images (*n =* 3). Data are presented as mean ± SD. * *p* < 0.05, ** *p* < 0.01, *** *p* < 0.001 compared to M0, ^#^ *p* < 0.05, ^##^ *p* < 0.01, and ^###^*p* < 0.001 compared to M0 + Exo. *n =* 3–4/group.

**Figure 5 pharmaceutics-15-01489-f005:**
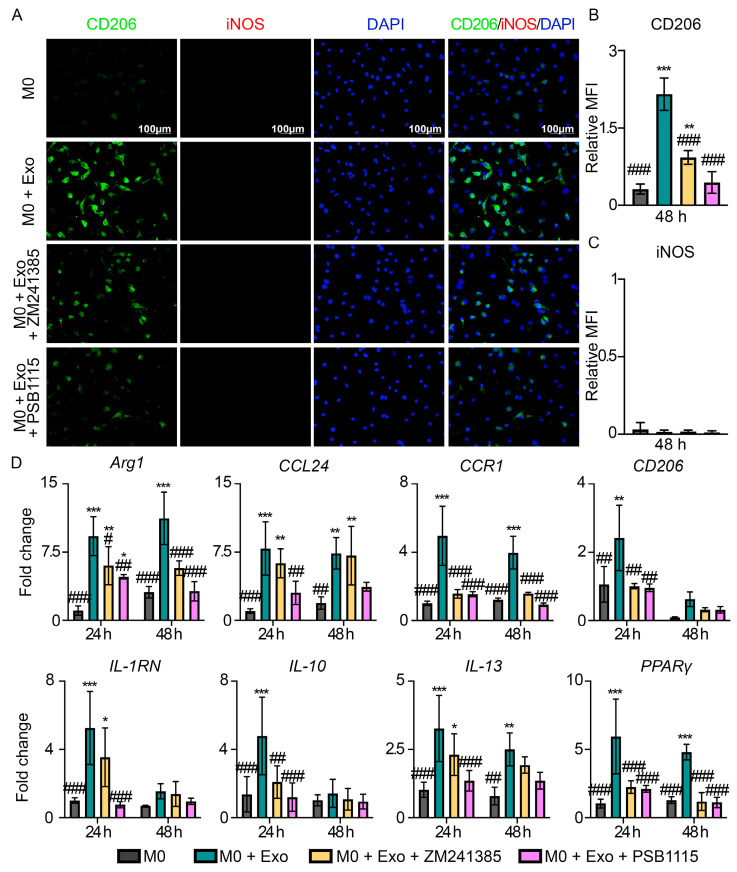
MSC exosomes modulate macrophage polarization through specific adenosine receptors A_2A_ and A_2B_. (**A**) IF staining at 48 h for CD206 and iNOS in primary macrophages treated with MSC exosomes, following pre-treatment with adenosine A_2A_ receptor inhibitor (ZM241385) and A_2B_ receptor inhibitor (PSB1115). Representative images (*n =* 3). Scale bar: 100 μm. Quantitative MFI of (**B**) CD206 and (**C**) iNOS at 48 h. (**D**) Gene expression analysis of M2-associated genes at 24 and 48 h. Data are presented as mean ± SD. * *p* < 0.05, ** *p* < 0.01, *** *p* < 0.001 compared to M0, ^#^ *p* < 0.05, ^##^ *p* < 0.01, and ^###^ *p* < 0.001 compared to M0 + Exo. *n =* 3–4/group.

**Figure 6 pharmaceutics-15-01489-f006:**
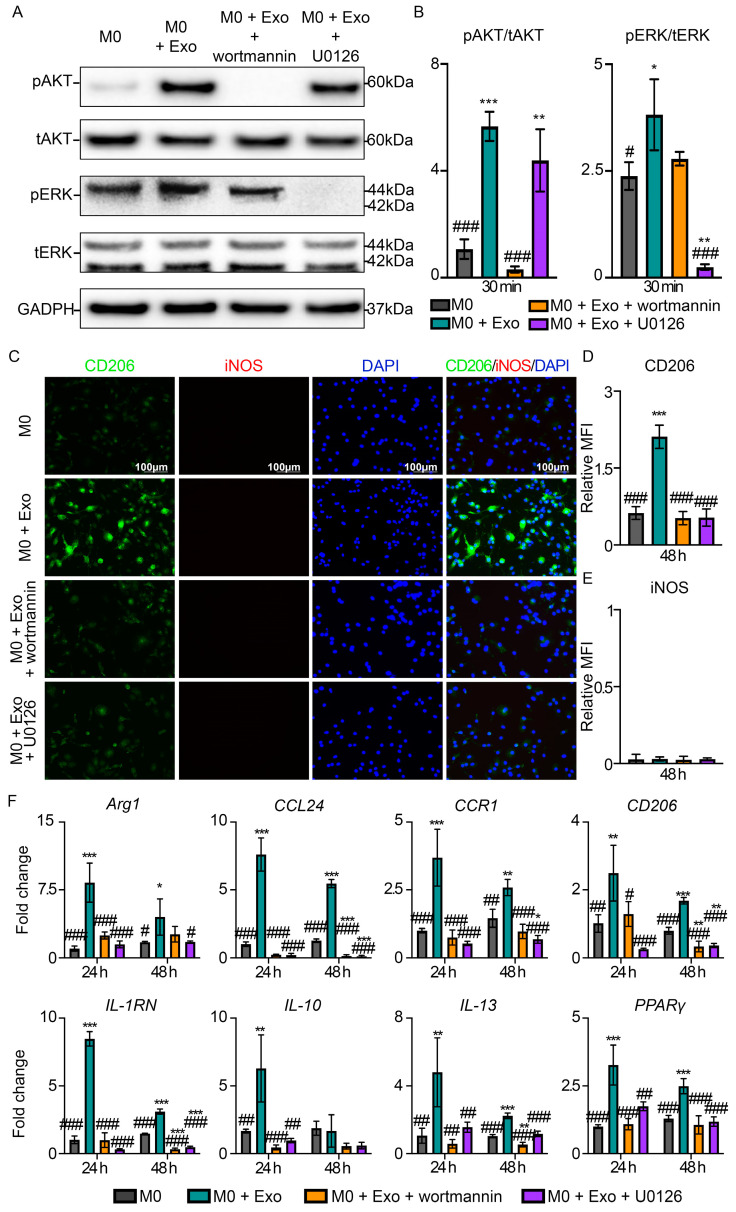
MSC exosomes promote M2 macrophage polarization via AKT and ERK signaling pathways. (**A**) Western blotting and (**B**) semi-quantitative analysis of AKT and ERK phosphorylation in primary macrophages pre-treated with AKT inhibitor (wortmannin) or ERK inhibitor (U0126) before exosome treatment. Representative images (*n =* 3). (**C**) IF staining at 48 h for CD206 and iNOS in primary macrophages treated with MSC exosomes, following pre-treatment with AKT inhibitor (wortmannin) and ERK inhibitor (U0126). Representative images (*n =* 3). Scale bar: 100 μm. Quantitative MFI of (**D**) CD206 and (**E**) iNOS at 48 h. (**F**) Gene expression analysis of M2-associated genes at 24 and 48 h. Data are presented as mean ± SD. * *p* < 0.05, ** *p* < 0.01, *** *p* < 0.001 compared to M0, ^#^ *p* < 0.05, ^##^ *p* < 0.01, and ^###^ *p* < 0.001 compared to M0 + Exo. *n =* 3–4/group.

**Figure 7 pharmaceutics-15-01489-f007:**
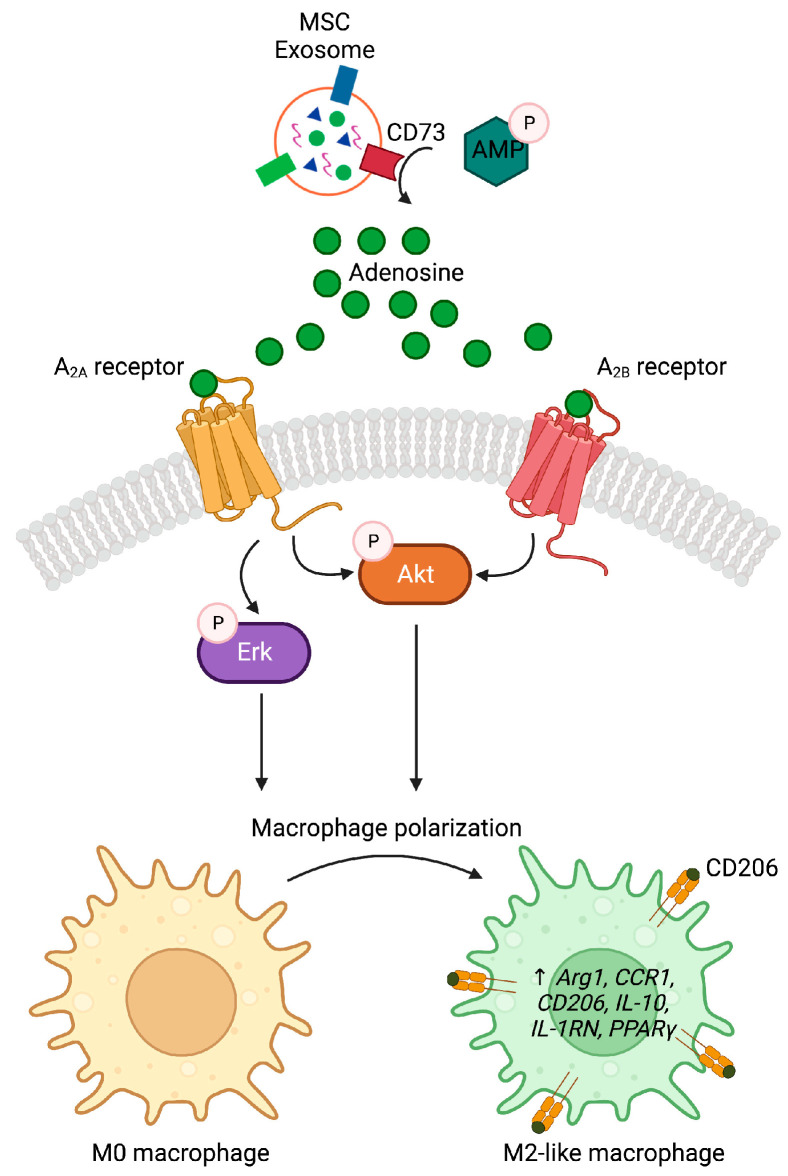
Proposed mechanism of MSC exosomes in promoting M2 macrophage polarization. Exosomal CD73 converts AMP to adenosine, which in turn interacts with A_2A_ receptor to activate AKT and ERK signaling, and with A_2B_ receptor to activate AKT signaling to promote M2 macrophage polarization. Image is created with Biorender.com on 8^th^ May 2023.

**Table 1 pharmaceutics-15-01489-t001:** Primer sequences.

Gene	Primer Sequences	
*Arg1*	F	TTGATGTTGATGGACTGGAC
	R	TCTCTGGCTTATGATTACCTTC
*CCL24*	F	CCAAGGCAGGGGTCATCTTC
	R	ACCTTGGTGCTATTGCCTCG
*CCL5*	F	CGTGAAGGAGTATTTTTACACCAGC
	R	CTTGAACCCACTTCTTCTCTGGG
*CCR1*	F	AAGTACCTTCGGCAGCTGTTTC
	R	ACAGAGAAGAAGGGCAGCCAT
*CD206*	F	GGGTTCACCTGGAGTGATGG
	R	ATTGTCTTGAGGAGCGGGTG
*CD80*	F	CCAAGTGTCCAAGTCGGTGA
	R	TTGTACTCGGGCCACACTTC
*GAPDH*	F	GGTCGGTGTGAACGGATTTGG
	R	GCCGTGGGTAGAGTCATACTGGAAC
*IFN-γ*	F	GATCCAGCACAAAGCTGTCA
	R	GACTCCTTTTCCGCTTCCTT
*IL-10*	F	CTGTCATCGATTTCTCCCCTGT
	R	CAGTAGATGCCGGGTGGTTC
*IL-1β*	F	CCTCTGACAGGCAACCACTT
	R	CATCCCATACACACGGACAA
*IL-12β*	F	CCGATGCCCCTGGAGAAAC
	R	CCTTCTTGTGGAGCAGCAG
*IL-13*	F	GCTTATTGAGGAGCTGAGCAACA
	R	GCCAGGTCCACACTCCATA
*IL-1RN*	F	GTGTTCTTGGGCATCCACGG
	R	TGTCTCGGAGCGGATGAAGG
*IL-6*	F	CCAGGTTCTCTTCAAGGGACAA
	R	GGTATGAAATGGCAAATCGGCT
*iNOS*	F	GAAACTTCTCAGCCACCTTGG
	R	CCGTGGGGCTTGTAGTTGAC
*PPARγ*	F	AGGATTCATGACCAGGGAGTT
	R	AGCAAACTCAAACTTAGGCTCCAT
*TNF-α*	F	CCAGGTTCTCTTCAAGGGACAA
	R	GGTATGAAATGGCAAATCGGCT

**Table 2 pharmaceutics-15-01489-t002:** List of antibodies.

Antibody	Supplier	Clone	Dilution
anti-CD73	GeneTex	Rabbit polyclonal	1:1000
anti-phospho-AKT (Ser473)	CST	D9E	1:1000
anti-AKT (pan)	CST	11E7	1:1000
anti-phospho-ERK1/2 (Thr202/Tyr204)	CST	Rabbit polyclonal	1:1000
anti-ERK1/2	CST	137F5	1:1000
anti-GAPDH	Abcam	6C5	1:10,000

## Data Availability

Data are available upon reasonable request to the corresponding author.

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
