# Peer review of "Mesenchymal Stromal Cell Exosomes Mediate M2-like Macrophage Polarization through CD73/Ecto-5′-Nucleotidase Activity"

_pharmaceutics, 2023, doi:10.3390/pharmaceutics15051489_

Round 1
Reviewer 1 Report
The manuscript by Teo et al. titled "Mesenchymal stromal cell exosomes mediate M2-like macrophage polarization through CD73/ecto-5' -nucleotidase activity" is an interesting work regarding MSC exosomes. The paper is well written and easy to follow because of the straight forward style and presentation. The figures are well put together and provide important and relevant information. Especially the methods figure and summary figure are extremely helpful for the reader. The manuscript would benefit from addressing the following minor points:
1) What is the rationale for using E1-MYC 16.3 ESC-derived MSCs for the studies? Why not other MSCs? How do these MSCs differ from MSCs that exist naturally in the body? Please explain in text and in the response.
2) Line 295, change "strongly indicate" to demonstrate or suggest.
3) Please include M2 macrophage characteristics observed in this study into Figure 7.
Reviewer 2 Report
The manuscript studied the role of Mesenchymal stromal cell exosomes in M2-like macrophage polarization. The authors suggested that MSC exosomes promote M2-like macrophage polarization by catalyzing the production of adenosine, which then binds to adenosine receptors A2A and A2B to activate AKT/ERK-dependent signaling pathways. Also, polarization of M2-like macrophage by MSC exosomes was abolished in the presence of inhibitors of CD73 activity, adenosine receptors A2A and A2B, and AKT/ERK phosphorylation.
My comments
- In figure 6a: total ERK bands are partially cropped, and in p-AKT band for M0+Exo+wortmannin the background is different, please comment
-The inflammatory cytokines expression assessed by PCR should be confirmed by protein measurement by ELISA
The manuscript needs revision for minor grammar and language mistakes
Round 2
Reviewer 2 Report
The authors addressed my comments
English language needs minor revision